# Effect of Surface Treatments and Thermal Aging on Bond Strength Between Veneering Resin and CAD/CAM Provisional Materials

**DOI:** 10.3390/polym17050563

**Published:** 2025-02-20

**Authors:** Ali Robaian, Abdullah Mohammed Alshehri, Nasser Raqe Alqhtani, Abdulellah Almudahi, Khalid K. Alanazi, Mohammed A. S. Abuelqomsan, Eman Mohamed Raffat, Ali Elkaffas, Qamar Hashem, Tarek Ahmed Soliman

**Affiliations:** 1Conservative Dental Science Department, College of Dentistry, Prince Sattam bin Abdulaziz University, Al-Kharj 11942, Saudi Arabia; ali.alqhtani@psau.edu.sa (A.R.); am.alshehry@psau.edu.sa (A.M.A.); a.almudahi@psau.edu.sa (A.A.); kk.alanazi@psau.edu.sa (K.K.A.); m.abuelqomsan@psau.edu.sa (M.A.S.A.); a.elkaffas@psau.edu.sa (A.E.); 2Department of Oral and Maxillofacial Surgery and Diagnostic Sciences, College of Dentistry, Prince Sattam bin Abdulaziz University, Al-Kharj 11942, Saudi Arabia; n.alqhtani@psau.edu.sa; 3Prosthetic Dentistry Department, Faculty of Dentistry, Benha National University, Benha 13511, Egypt; emanraffat70@gmail.com; 4Prosthetic Dentistry Department, General Zagazig Hospital, El Sharkeya 44516, Egypt; 5Operative Dentistry Department, Faculty of Dentistry, Mansoura University, Mansoura 35516, Egypt; 6Endodontic division-Conservative Dental Science Department, College of Dentistry, Prince Sattam bin Abdulaziz University, Al-Kharj 11942, Saudi Arabia; q.hashem@psau.edu.sa

**Keywords:** aging, bond strength, provisional restoration, surface treatments

## Abstract

The oral environment significantly influences the esthetic appearance of CAD/CAM provisional restorative materials. Therefore, a veneering layer is required. Bonding veneering resin composites to these materials presents challenges, particularly under conditions of thermal aging. This study evaluated the impact of various surface treatments and thermal aging on the bond strength between veneering resin and CAD/CAM provisional materials. Fifty disk-shaped specimens of each CAD/CAM material (CAD-Temp, Everest C-Temp, and PEEK), measuring 10 mm in diameter and 3 mm in height, were fabricated. After being ultrasonically cleaned, specimens were embedded in acrylic resin blocks, leaving one surface exposed for surface treatments. Specimens were assigned to five groups at random. Group C: no surface treatments applied; DB: mechanically roughened with a diamond bur; DB + TC: DB group subjected to 5000 cycles of thermocycling; SB: treated with aluminum oxide airborne abrasion; SB + TC: SB group subjected to 5000 cycles of thermocycling. After the surface treatments, the primer and resin veneering composite were applied to the specimens. The shear bond strength (SBS) was calculated using a universal testing machine and the mode of failure was evaluated with an optical stereomicroscope with 40× magnification. Scanning electron microscopy evaluation was conducted to examine the surface topography of the materials’ surfaces after surface treatments. C-Temp in the SB group exhibited the highest SBS values (20.38 ± 1.04 MPa), while CAD-Temp in the C group showed the lowest values (4.60 ± 0.54 MPa). PEEK recorded significantly higher SBS values in DB + TC and SB + TC groups (9.26 ± 1.07 and 12.92 ± 0.97 MPa, respectively) compared to CAD-Temp in DB + TC and SB + TC groups (6.04 ± 0.76 and 8.82 ± 0.86 MPa, respectively). C-Temp exhibited higher SBS without surface treatment (13.11± 0.55 MPa), whereas PEEK showed higher SBS after diamond bur roughening and air particle abrasion (10.87 ± 1.02 MPa, and 14.37 ± 0.98 MPa, respectively). The thermocycling significantly reduced SBS values for C-Temp in the DB + TC and SB + TC groups (11.18 ± 0.92, 15.56 ± 0.87 MPa, respectively) and CAD-Temp in the DB + TC and SB + TC (6.04 ± 0.76 MPa and 8.82 ± 0.86 MPa, respectively). Conversely, the thermocycling had no significant effect on SBS values for PEEK material in the air particle abrasion group (12.92 ± 0.97 MPa).

## 1. Introduction

Provisional restorations are frequently used in fixed partial denture (FPD) treatment. Before placing the permanent dental prosthesis, they ensure the stability of the occlusal relationships and assess the treatment’s efficacy in terms of esthetic, functional, and therapeutic advantages [1]. Recent advancements in CAD/CAM technology have enhanced the fabrication of provisional restorations via indirect methods. Resin CAD/CAM provisional materials are industrially polymerized blocks under high pressure and temperature, resulting in a higher degree of conversion and polymerization reaction than conventional polymerized resins [2,3,4]. These conditions provide provisional restorations that exhibit superior mechanical characteristics compared to those fabricated manually. Their excellent mechanical properties serve as an effective solution for durable provisional restorations. Furthermore, the enhanced fit of milled CAD/CAM restorations is expected to reduce the chances of bacterial contamination and protect the pulp from harmful temperature fluctuations [5,6]. These restorations are placed in the oral cavity for prolonged durations in some clinical scenarios, including adjustments to the vertical dimension, modifications of the occlusal plane, and crown lengthening procedures. Conversely, manual fabrication is associated with several issues, such as undesirable odor, significant polymerization shrinkage, reduced durability, porosity, and increased surface roughness [7,8]. Compared to the traditional approach, they have reduced the number of workflow steps and required working time.

Over the past few years, the role of provisional restorations has undergone significant transformation. These restorations are now considered provisional with specific functions and purposes rather than temporary restorations. Provisional restorations are a crucial issue when comprehensive occlusal reconstruction is required, especially when subjected to prolonged functional stresses. Furthermore, they may provide new therapeutic options in maxillofacial rehabilitation, implant-assisted prostheses, and periodontal therapy [1,2]. Provisional restorations serve as diagnostic tools to assess the positioning and contours of the intended definitive restoration during the healing phase. Their role is to facilitate the healing of peri-implant tissue and enable the clinician to assess any required phonetic or esthetic modifications. Provisional implant restorations enable patients to visualize and assess the final restorative outcome, facilitating acceptance and guiding necessary modifications for definitive restoration [9].

When choosing long-term provisional restorative materials, clinicians should consider the type of materials, their mechanical and bonding properties, and the impact of oral environmental conditions when using provisional CAD/CAM materials for extended periods [10,11,12]. Although these restorations demonstrate significant mechanical potential, resin veneering is essential for esthetic enhancement. Veneering of CAD/CAM provisional restorations is a necessary step to improve the esthetic outcomes of restorations, as indicated by the manufacturers. Furthermore, most polymeric CAD/CAM blocks are available in a single-color shade. For long-term applications of these polymeric materials, the use of veneering resins is crucial to enhance esthetic results [13,14]. Additionally, these materials are frequently subjected to oral conditions, including rapid temperature fluctuations. Thermal stress can cause deterioration of the interfacial bonding when hot or cold beverages are consumed in the mouth [13,15,16].

Bonding CAD/CAM provisional restorations to resin veneering is a difficult challenge because the resin is resistant to surface modification due to its high degree of polymerization and diverse microstructures. The establishment of a suitable bond between the veneering resin and the CAD/CAM provisional materials is essential for the success of these restorations [17,18,19]. Weigand et al. [17] assessed the bond strength of various CAD/CAM polymers to resin composites following surface treatments. The control group without surface treatment exhibited shear bond strength values between 1.1 and 9.4 MPa. The bur roughening group showed values ranging from 8 to 21 MPa, while the air particle abrasion group displayed values from 11 to 20 MPa, depending on the type of material utilized. The study concluded that mechanical surface treatment is a critical step, as it substantially improves adhesion. Furthermore, Gouveia et al. [19] evaluated the influence of surface treatments on the shear bond strength of veneering composite resin to milled PEEK. The study concluded that PEEK surfaces subjected to aluminum oxide airborne particle abrasion exhibited superior shear bond strength (15 MPa) when bonded to composite resin veneering. Yao et al. [15] assessed the flexural strength of CAD/CAM interim materials before and following thermal cycling. The flexural strength values for CAD-Temp decreased from 96.84 ± 3.82 MPa to 77.27 ± 7.02 MPa. These findings were attributed to the effects of thermal changes due to thermocycling and the reduced filler content within the materials.

It is crucial to have a comprehensive understanding of the bonding characteristics of these materials, especially when exposed to thermal aging. This study investigated the impact of different surface treatments and thermal aging on the bond strength between veneering resin and CAD/CAM provisional restorative materials. The null hypotheses examined were: (1) there are no significant differences in bond strength among the three CAD/CAM materials, (2) there are no significant differences between the different surface treatment methods in enhancing the bond strength, and (3) there is no significant impact of thermocycling on the bond strength of these materials.

## 2. Materials and Methods

Table 1 shows the materials utilized in this study. CAD/CAM polymers used for long-term provisional restorations are utilized in this study, including polyacrylate polymer (CAD-Temp, VITA Zahnfabric, Germany; CAT), fiberglass-reinforced polymer (Everest C-Temp, KaVo, Biberach, Germany; CT), and polyether ether ketone PEEK (BioHPP, Bredent, Germany; PK). The sample size calculation was performed using G* Power software. The power analysis indicated that ten specimens per group were necessary to achieve a 0.95 power at a 5% significance level (effect size = 0.385, α = 0.05 for a two-tailed test).

### 2.1. Specimens’ Preparation and Grouping

Fifty disk-shaped specimens of each CAD/CAM material (measuring 10 mm diameter × 3 mm height) were fabricated by sectioning the block using an ISOMET (Techcut4, Allied, Rancho Dominguez, CA, USA). Then, the specimens were ultrasonically cleaned using 90% isopropyl to enhance the removal of foreign contaminants from surfaces. To prepare the cut surfaces for resin veneers, wet silicon carbide polishing papers (Microcut™, Buehler, Lake Bluff, IL, USA) of different grades (600, 800, and 1200 grit) were utilized. A uniform finish was achieved using polishing paste (Diamat, Pace Technologies, Tuscon, AZ, USA) and a 1 μm polishing cloth disk (Grinder polisher Metaserve 250; BUEHLER, Lake Bluff, IL, USA). The specimens were embedded in acrylic resin blocks (Paladur, Heraeus-Kulzer, Hanau, Germany), with one surface left exposed for surface treatments.

Specimens were categorized into 5 groups (n = 10/gp) based on the surface treatments and aging procedures outlined as follows. C: no surface treatment. DB: the surfaces of the specimens were abraded with a diamond bur (medium grit) that was mounted in a high-speed handpiece operating at 45,000 rpm, with irrigation, for 8 s [14]. DB + TC: DB group with 5000 cycles of thermocycling. SB: aluminum oxide particles (50 μm, LEMAT NT4, Wassermann, Germany) were used to air-abrade the specimens’ surfaces for ten seconds, maintaining a distance of 10 mm and applying a 0.55 MPa pressure. This was followed by air drying the surfaces for 20 s [20,21,22]. SB + TC: SB group, along with 5000 cycles of thermocycling [23]. The study design is shown in Figure 1.

### 2.2. Bonding Procedures and Aging Protocol

The bonding region was defined by attaching a 6 mm diameter double-sided tape with a circular hole to the specimens’ surfaces. The primer (Visio. Link, Bredent GmbH & Co., Senden, Germany) was applied to the specimens’ surfaces according to the manufacturer’s instructions and cured for 20 s using an LED light (Elipar Freeligh 2, 3M ESPE, 1226 mW/cm^2^). Incremental packing of resin veneering materials (Crealign paste, Bredent GmbH & Co., Senden, Germany) was applied onto the treated CAD/CAM surfaces, with each layer (2 mm) cured using LED light for 180 s according to manufacturer instructions to produce a 10 mm height cylinder [24]. In each group, half of the specimens (N = 10) underwent 5000 cycles of thermocycling (SD Mechatronic GmbH, Feldkirchen Westerham, Germany) at temperatures ranging from 5 to 55 °C for 30 s.

### 2.3. Shear Bond Strength (SBS) Testing

The SBS between the resin veneer and CAD/CAM provisional materials was calculated using a universal testing machine. The machine’s lower jaw firmly clamped the specimens, ensuring their alignment with the direction of the shear force. The specimen underwent a compressive load at a crosshead speed of 0.5 mm/min until failure, with the load being recorded via a force gauge [4,18]. For each specimen, the maximum load was divided by the surface area to obtain the SBS in MPa. The failure mode was classified as an adhesive failure occurring between the resin veneering and the provisional material surface, cohesive failure within either the resin veneering or the provisional material surface, or mixed failure, i.e., both adhesive and cohesive failure. A frequency analysis was performed for each failure mode utilizing an optical stereomicroscope at a magnification level of 40×.

### 2.4. Scanning Electron Microscopy (SEM)

A scanning electron microscopy (SEM) instrument (Jeol-JSM-6510, Tokyo, Japan) was used to identify surface differences in the CAD/CAM provisional restorative materials following surface treatments. All specimens were coated using a gold sputter coater. SEM analysis was performed for each group at a magnification of 1000× [23].

### 2.5. Statistical Analysis

The Shapiro–Wilk test assessed the normality of the data distribution, while Levene’s test evaluated the homogeneity of variances. All SBS values conformed to normality and satisfied the assumption of homogeneity of variances. A three-way analysis of variance (ANOVA) with Bonferroni’s post hoc test was performed at a 95% confidence level (α = 0.05) to evaluate the main effects of each material, surface treatment, and thermocycling, as well as their interaction effect on the shear bond strength (SBS).

## 3. Results

Table 2 presents the SBS values (MPa) for each group. The three-way analysis of variance showed significant differences in SBS values among different types of materials (F = 813.74, *p* < 0.001), surface treatments (F = 602.203, *p* < 0.001), and the aging conditions (F = 244.167, *p* < 0.001) [Table 3]. The highest SBS values were observed for the C-Temp in the SB group (20.38 ± 1.04 MPa), and the lowest values were noted for the CAD-Temp in the C group (4.60 ± 0.54). The DB and SB groups recorded the highest significance (*p* < 0.001) SBS values in C-Temp (16.92 ± 0.70, 20.38 ± 1.04 MPa, respectively) compared to other surface treatment groups in different materials.

No significant differences were observed in SBS values between CAD-Temp and PEEK in the C, DB, and SB groups (*p* > 0.05). On the other hand, PEEK recorded significantly higher SBS values in the DB + TC and SB + TC groups (9.26 ± 1.07, 12.92 ± 0.97 MPa, respectively) compared to CAD-Temp in the DB + TC and SB + TC groups (6.04 ± 0.76, 8.82 ± 0.86 MPa, respectively). Thermocycling significantly reduced SBS values for C-Temp in the DB + TC and SB + TC groups (11.18 ± 0.92, 15.56 ± 0.87 MPa, respectively) and for CAD-Temp in the DB + TC and SB + TC (6.04 ± 0.76 MPa and 8.82 ± 0.86 MPa, respectively). Conversely, thermocycling had no significant effect on SBS values for PEEK material in the air particle abrasion group (12.92 ± 0.97 MPa).

Adhesive failure was highly prevalent in CAD-TEMP in all groups. On the other hand, mixed failures were the prevalent type for C-Temp in all groups. Regarding PEEK material, adhesive failure was the prevalent type observed in group C. However, mixed failure was the most common type in the SB group. Following thermocycling, the most common failure types observed were mixed and cohesive in C-temp and PEEK, whereas adhesive failure was the most prevalent in CAD-Temp (Figure 2).

The SEM analysis indicated differences in the surface microstructures of the treated CAD-Temp, C-Temp, and PEEK (Figure 3). CAD-Temp and PEEK materials in the C group showed smooth surfaces devoid of surface texture, whereas C-Temp showed homogenous, smooth surfaces with irregular surface texture (Figure 3A,D,G). Roughening using a bur consistently showed an erosive appearance with undercuts (Figure 3B,E,H). The airborne-abraded group exhibited clearly defined micro-sized irregularities (Figure 3C,F,I). The impact of mechanical roughening methods, such as diamond bur and sandblasting, exhibited greater homogeneity, uniformity, and orientation with C-Temp.

## 4. Discussion

The three-way ANOVA indicates a significant impact of the three independent variables (material type, surface treatment methods, and thermal aging) on the SBS. Therefore, the three null hypotheses are rejected.

Provisional restorations serve as temporary treatment to maintain occlusion, pulp health, and esthetic appearance while permanent restorations are being fabricated. Provisional restorations can be fabricated manually or by CAD/CAM technology. The establishment of a suitable bond between the veneering resin and the CAD/CAM provisional materials is essential for the success of these restorations. In this study, three independent variables were investigated (type of material, surface treatments, and thermal aging). Three different CAD/CAM materials for long-term provisional restoration were investigated. Due to the diverse microstructure, their response may vary to surface treatments and consequently affect bond strength. Furthermore, the polymeric CAD/CAM materials utilized in this study have been recommended by manufacturers as suitable framework materials for implant-supported fixed prostheses [25,26,27]. For long-term application, these materials may be veneered post-milling through a layering technique to improve esthetic outcomes. The bond strength of veneering resin to CAD/CAM provisional materials must be sufficiently strong to enhance their durability in oral environments during treatment procedures. This investigation evaluates the influence of surface treatments and aging on the bond strength between veneering resin composites and long-term CAD/CAM provisional restorative materials.

Multiple surface treatment techniques can be utilized to improve the bond strength of interim restorative materials, such as air abrasion, laser treatment, and acid etching [28,29]. Airborne particle abrasion can be used as a surface treatment method. It has been reported that it is the easiest way to improve microroughness and increase the surface area of polymer-based dental materials for sufficient bonding [4,18,23]. Utilizing rotary instruments as a roughening approach offers a simple and cost-effective solution while also simulating clinical use in practice to enhance microroughness and augment surface area for optimal bonding [30,31]. There are two types of instruments: tungsten carbide or diamond bur. Tungsten carbide is a cutting instrument with defined blades, whereas the diamond bur is an abrasive instrument and has geometrically undefined grains. In the literature, there is controversy regarding the performance of diamond or carbide burs. In this study, roughening with a diamond bur was utilized according to previous studies that indicated the use of a diamond bur as a surface treatment method for effectively enhancing bonding [4,18,23,32,33,34].This study conducted shear bond strength tests, which serve as a reliable method for evaluating bond strengths of a large surface area, typically between 3 and 6 mm in diameter [4,19,35].

This study’s results indicate that the untreated surfaces of CAD-Temp and PEEK displayed the lowest SBS values. This is likely due to these materials being industrially polymerized, resulting in higher polymerization and an insufficient presence of free radicals for effective adhesion to the resin veneering materials [14,36]. On the other hand, the untreated surfaces of C-Temp showed the highest SBS in the control group. The irregular surface topography of C-Temp (Figure 3D) may facilitate adhesive resin penetration, thereby enhancing the interlock with resin veneering materials [7,17]. These findings align with previous studies [4,17], which proposed that the increased SBS with C-Temp may result from the adhesive’s capacity to infiltrate surface irregularities associated with glass fiber, thereby enhancing bonding.

This study demonstrates that mechanical surface treatments utilizing a diamond bur and air particle abrasion lead to a significant enhancement in shear bond strength (SBS), especially in C-Temp and PEEK materials. Shear bond strength (SBS) can be enhanced through mechanical surface treatments, which increase the substrate’s surface energy and produce irregularities that aid micromechanical retention. These findings are consistent with previous studies that reported that the mechanical surface treatment utilizing diamond burs and air particle abrasion enhanced adhesion [18,23,30,31]. Additionally, C-Temp exhibits a higher glass fiber content and is classified as a high-performance continuous molecular plastic polymer chain, which is appropriate for adhesive resin penetration [4,17]. The findings align with Weigand et al. [17], who proposed that the increased SBS with C-Temp may result from the adhesive’s capacity to infiltrate surface irregularities associated with glass fiber, thereby enhancing adhesion. The SEM analysis in this study demonstrated that mechanical roughening modified the surface morphology of PEEK, enhancing adhesive resin penetration, which improved micromechanical interlock and subsequently increased bond strength.

Thermal cycling is extensively used to simulate the constantly changing temperatures in the oral environment. The specimens were thermocycled for 5000 cycles at temperatures ranging from 5 °C to 55 °C, with a dwell duration of 30 s to simulate aging. A total of 5000 cycles were used to approximate six months of clinical service in the oral cavity [37]. The current investigation revealed that thermocycling significantly reduced the SBS of CAD-Temp. The high polymeric content (83–86 wt% PMMA) may account for this phenomenon, as it is prone to water infiltration between the gaps of the polymer chains, resulting in their separation. This process leads to water absorption, which ultimately softens the resin matrix and adversely affects SBS [4,16,23]. Thermocycling led to a reduction in the shear bond strength (SBS) of C-Temp. The presence of moisture promotes the corrosion of the glass fiber surface, as water infiltrates the polymer matrix, thereby compromising mechanical properties and bond strength [38]. In contrast, thermocycling did not significantly affect the SBS of PEEK material, likely due to its low water sorption. PEEK demonstrates a water sorption value of (≤6.5 μg/mm^3^), CAD-Temp presents a value of (≤40 μg/mm^3^), and Everest C-Temp has a value of (9.6 μg/mm^3^) [8,23]. A previous investigation indicated that 5000 thermocycling cycles had a minimal impact on the adhesion characteristics of PEEK restorative materials [39]. Libermann et al. [18] investigated the impact of different aging conditions on the water sorption characteristics of multiple CAD/CAM polymers. The findings indicated that the storage media type did not have a significant effect on PEEK’s water absorption capacity.

This study showed that C-Temp and PEEK demonstrated a higher frequency of mixed failures, which can be ascribed to the inconsistent distribution of shear forces at the resin-restoration interface [40,41]. Furthermore, a significant occurrence of adhesive failures at CAD-Temp was observed, due to inadequate SBS that maximizes the adhesive failure. The results demonstrate a shift from adhesive failure to mixed failure as the bond strength increases. Conforming to the requirements of ISO 10,477 [42], the minimal acceptable SBS value at the interface of resin-based materials with the substrate is 5 MPa. A clinically acceptable SBS value of 10 MPa was proposed by Beher et al. [43]. All groups met the clinical criteria, except for the C and SB-T groups in CAD-Temp, the SB-T group in C-Temp, and the C group in PEEK.

This study has limitations, as it utilizes only three types of provisional restorative materials and one type of veneering resin. Furthermore, other oral environmental factors, such as different pH levels and prolonged aging periods, need further investigation. SBS values obtained in this study serve as relative comparisons between the tested materials and do not fully represent the entire intraoral forces in the oral environment. Furthermore, shear bond strength values exhibit significant variability based on study design; therefore, caution is warranted when translating laboratory bond strength findings to clinical applications. From a material science perspective, the reason for the differences in bonding performance among these restorative materials could be related to differences in elastic modulus. Consequently, future research should consider the impact of the elastic modulus on the bonding capabilities of these materials. Further investigation should consider color stability, especially at the interface between the veneering layer and CAD/CAM provisional materials. Additionally, further characterization analysis should be performed on the bonding interface between the CAD/CAM provisional materials and the resin veneering.

## 5. Conclusions

Within the limitations of this study, the following can be concluded:C-Temp exhibited higher SBS values without surface treatment (13.11 ± 0.55 MPa), whereas PEEK showed higher SBS values after diamond bur roughening and air particle abrasion (10.87 ± 1.02 MPa, and 14.37 ± 0.98 MPa, respectively);CAD-Temp recorded the lowest SBS values, which are below the clinically accepted value (10 MPa) in all groups except for the SB group (12.86 ± 0.75 MPa);Thermocycling significantly reduced bond strength in all materials except for PEEK material in the air particle abrasion group (12.92 ± 0.97 MPa);The results of the current study suggest that C-Temp and PEEK can be recommended as promising provisional materials due to their appropriate bond strength values after thermal aging. Suitable surface treatments and the selection of suitable provisional material could improve the adhesive properties for these materials to be used for long-term applications.

## Figures and Tables

**Figure 1 polymers-17-00563-f001:**
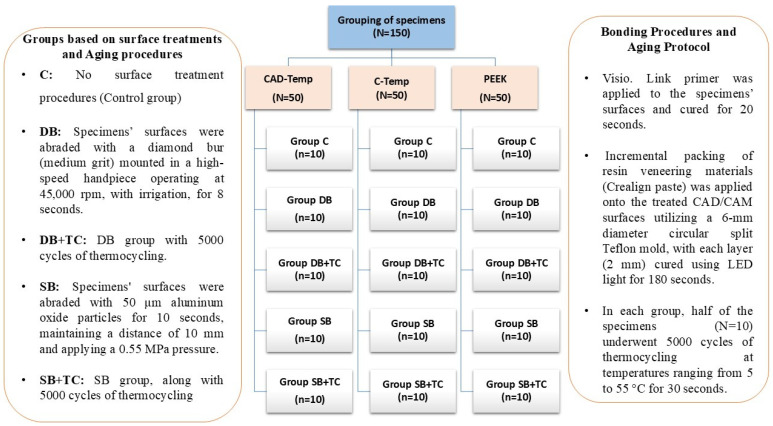
Study design and specimens’ grouping.

**Figure 2 polymers-17-00563-f002:**
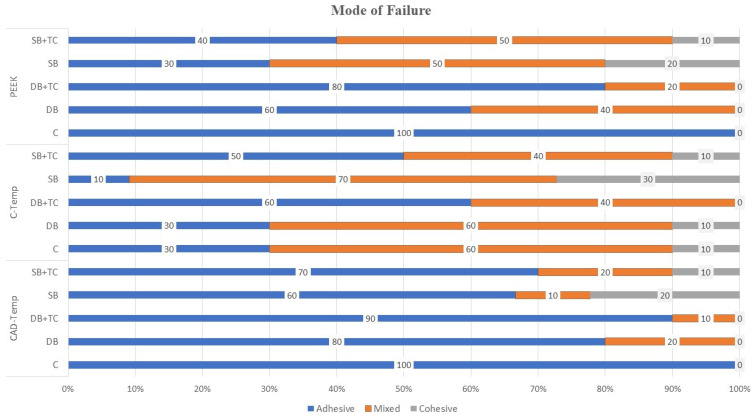
Mode of failure for all groups.

**Figure 3 polymers-17-00563-f003:**
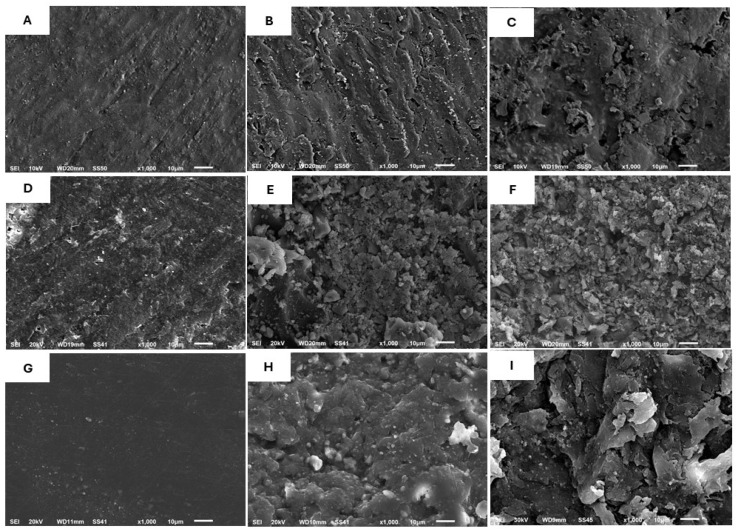
SEM micrographs (1000): (**A**) CAD-TEMP control group, (**B**) CAD-TEMP bur roughening, (**C**) CAD-TEMP—air abrasion, (**D**) C-TEMP control group, (**E**) C-TEMP bur roughening, (**F**) C-TEMP—air abrasion, (**G**) PEEK Control group, (**H**) PEEK bur roughening, and (**I**) PEEK—air abrasion.

**Table 1 polymers-17-00563-t001:** Materials used in the study.

Product	Composition/Manufacturer	Indication	Lot. No.
CAD-Temp	- 83–86 wt% PMMA, 14 wt% micro filler (silica), Pigments (<0.1%). - VITA Zahnfabrik	Multi-unit, fully or partially anatomical long-term temporary bridges with up to 2 pontics.	38590
Everest C-Temp	- Fiberglass-reinforced polymer. - High-performance endless molecular polymer chain plastic. - KaVo, Biberach, Germany.	Long-term temporary restoration up to 6 units.	6946
Bre CAM Bio HPP	- Polyether ether ketone, 20 wt% titanium dioxide ceramic filler and aluminum oxide sand (50 µm mean particle size). - Bredent GmbH & Co., Senden, Germany.	4-part posterior bridge up to two pontics.	56654456
Visio. Link Primer	- MMA, PETIA, photoinitiators - Bredent GmbH & Co., Senden, Germany.	Universal, light-curing PMMA and composite primer.	153141
Crea.lign Dentine Paste	- Acrylate oligomers, silanized inorganic fillers (50 wt% opalescent ceramic fillers), catalysts, and color pigments. - Bredent GmbH & Co., Senden, Germany.	Permanent resin veneering material.	N160407

**Table 2 polymers-17-00563-t002:** Shear bond strength (MPa) data (Mean ± SD) in all groups.

Groups	Provisional Restorative Materials		
	CAD Temp Mean ± SD	C-Temp Mean ± SD	PEEK Mean ± SD
C	4.60 ± 0.54 ^Bc^	13.11± 0.55 ^Ac^	5.9 ± 0.55 ^Bd^
DB	8.84 ± 0.55 ^Bb^	16.92 ± 0.70 ^Ab^	10.87 ± 1.02 ^Bb^
DB + TC	6.04 ± 0.76 ^Cc^	11.18 ± 0.92 ^Ac^	8.26 ± 1.07 ^Bc^
SB	12.86 ± 0.75 ^Ba^	20.38 ± 1.04 ^Aa^	14.37 ± 0.98 ^Ba^
SB + TC	8.82 ± 0.86 ^Bb^	15.56 ± 0.87 ^Ab^	12.92 ± 0.97 ^Aa^

Different upper-case letters in the same row indicate a significant difference between groups (Tukey’s test, *p* < 0.05). Different lower-case letters in the same column indicate a significant difference between groups (Tukey’s test, *p* < 0.05).

**Table 3 polymers-17-00563-t003:** Three-way ANOVA table for shear bond strength.

Source of Variations	Sum of Squares	df	Mean Squares	F	*p* Value
Corrected model	2719.531	14	194.252	279.932	*p* < 0.001
Intercept	15,187.369	1	15,187.369	21,886.146	*p* < 0.001
Materials	986.794	2	493.397	711.023	*p* < 0.001
Surface treatments	1173.509	2	586.754	845.557	*p* < 0.001
Aging	507.174	1	507.174	730.876	*p* < 0.001
Material × surface treatment	36.263	4	9.066	13.064	*p* = 0.001
Material × aging	153.790	2	76.895	110.811	*p* = 0.004
Surface treatment × aging	9.633	1	9.633	13.882	*p* < 0.001
Material × surface treatment × aging	4.860	2		3.502	*p* = 0.033
Error	93.680	135	2.430		
Total	20,679.429	150	0.694		
Corrected total	2813.211	149			

Statistically significant difference at *p* < 0.05.

## Data Availability

This article has all the data that were collected or analyzed during this study.

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
