# Peer review of "Effect of Surface Treatments and Thermal Aging on Bond Strength Between Veneering Resin and CAD/CAM Provisional Materials"

_polymers, 2025, doi:10.3390/polym17050563_

Round 1
Reviewer 1 Report
Comments and Suggestions for Authors
The surfaces of the spec- imens were abraded with a diamond bur (medium grit) that was mounted in a high-speed handpiece operating at 45,000 rpm, with irrigation, for 8 seconds[14] - Why were diamond drills used? Wouldn't it be better to use steel or carbide drills to obtain surface roughness?
Reviewer 2 Report
Comments and Suggestions for Authors
1) Abstract of the manuscript should be provided quantitative information in the process as well as outcome of the results for the better understanding of the readers. Briefly write in the abstract the methodology used to obtain the samples. Also highlight the best results obtained.
2) On the basis of plan of work and explanation of results, this manuscript can be considered as short-communication instead of full length article.
3) Quantitative information should be provided in the Introduction section.
4) Why this study is novel and different from other already published work?
5) Section 2: Material and Methods: This section about the procedure of work from “Preparation and Grouping of specimen” including “Bonding Procedures and Aging Protocol and bond strength evaluation” so much confusing. It will be better for readers if all procedures of adopted approaches and their analysis as per expected characterization techniques in different groups will be provided via schematic diagram along with conditions and compositions of the each combination.
6) Characterization analysis of the results about effect of aging, bonding procedure, bind strength testing and only SEM analysis is not appropriate for a scientific explanation of a research outcome. This section should be fully revised with the scientific proofs of other characterization analysis including X-ray photoelectron spectroscopy, FTIR and Raman analysis for the appropriate explanation.
7) The provide SEM Images are so poor. Every images should be in high resolution along with scale bar and scientific output of results.
8) The proper reason and scientific explanation is missing in the manuscript behind the bond strength variation, by effect of thermal aging and surface treatments.
9) The manuscript urgently needs linguistic revision due to containing many typo error.
10) There are many abbreviated words in this manuscript that need a full description. Please provide full description of terms in a separate unit. Fix that.
11) Conclusion: This part is very weak and completely unacceptable due to without quantitative evidence on the importance, outcomes and conclusions.
12) English of the manuscript should be revised carefully. There are several issues throughout the manuscript.
Comments on the Quality of English Language
English of the manuscript should be revised carefully. There are several issues throughout the manuscript.
Author Response
Dear reviewer
Your time and effort for these meticulous comments are highly appreciated. All the comments were checked and corrected throughout the manuscript. Kindly find the comments with confirmation of the correction at the attached file.

Reviewer 3 Report
Comments and Suggestions for Authors
Abstract:
1. There seems to not be a consistant use of terms. Is it CAD/CAM provisional restorations' or 'resin composites'.
2. Line 28 and 29 there is an extra space.
3. Should PK be PEEK?
Introduction
1. Line 46-49 - What about there role in patient acceptance of aesthetics of the final restoration or their ability to form soft tissue ie to develop an emergence profile during implant integration?
2. if you have three Null hypothesis please write them out completely so it is clear.
Method
1. It is unclear that materials used and how they we procured. Is CAD-Temp subtractively or additively manufactured? Is Everest C-Temp your control and thus a comparison of a conventional material. I am assuming that the BioHPP was subtractivly manufactured and this is why the narrative is around CAD/CAM materials?
2. Who are the manufacturers of the materials? ie BioHPP; PK = Bredent. Please list this correctly.
3. List in detail the manufacturing process of the samples prior to sectioning.
4. Why were the samples cleaned in IPA after sectioning - please justify.
5. List the polishing paste brand and manufacturer.
6. Why was 5000 cycles of thermocycling used - you need to justify in relation to the intended use of these materials.
7. Line 114 what is the abbreviation SBS?
8. Line 117 extra space and line 118 you seem to repeat yourself.
Results
1. Line 140 should this not be three-way analysis not two-way analysis
Discussion
1. At the start of the discussion you need to state if the Null hypothesis are accepted or rejected.
2. Line 187-189 - should be moved to the introduction.
3. The second paragraph is also part of the introduction. Is it relevant - do any of the materials you use int he study consist of PMMA?
4. Line 2.3 check spelling.
Author Response

(The authors gave the same response as above.)

Reviewer 4 Report
Comments and Suggestions for Authors
This study aimed at evaluating the bonding effect of veneering resin on CAD/CAM provisional resin composite materials. Three types of provisional RC materials were subjected to 2 different surface treatment, and bonding performance with and without artificial aging (thermal cycling) was tested.
The introduction is well-constructed and clear, providing adequate info regarding the study objective.
2.2 Did you use the bonding agent before incremental packing? Is this recommended to not using the bonding agent?
2.3 Please further define the identification of failure modes. Is there a percentage border among adhesive, cohesive, and mixed failure (such as 1/3 — 2/3, or 25% - 75% ?)
Table 2 & Fig 1: I will say the SBS of CAD Temp was not very high, and from the failure mode analysis, the adhesive failure is dominant. How will you interpret the result? Is resin veneering on this CAD/CAM material reliable and applicable to chairside?
typo on line 203
May further investigate colour stability, especially the interface between the veneering layer and and CAD/CAM RC materials.
Author Response

(The authors gave the same response as above.)

Round 2
Reviewer 2 Report
Comments and Suggestions for Authors
In this revised version, the authors improved the manuscript in terms of proper explanation, schematic diagram of study design (fig.1), and results and discussion with proper scientific explanation. Thus the present form of the manuscript can be accepted for publication.
However, in this manuscript number of the authors is around 10, how this is justified with respect to work of the manuscript?
Reviewer 3 Report
Comments and Suggestions for Authors
Thank you for making the suggested changes the manuscript is much improved and the concerns raised have been addressed